# BipA exerts temperature-dependent translational control of biofilm-associated colony morphology in *Vibrio cholerae*

Teresa del Peso Santos[1], Laura Alvarez[1], Brandon Sit[2], Oihane Irazoki[1], Jonathon Blake[3], Benjamin R Warner[4,5], Alyson R Warr[2], Anju Bala[1], Vladimir Benes[3], Matthew K Waldor[2], Kurt Fredrick[4,5], Felipe Cava[1]*

[1]The laboratory for Molecular Infection Medicine Sweden (MIMS), Department of Molecular Biology, Umeå University, Umeå, Sweden; [2]Howard Hughes Medical Institute, Brigham and Women's Hospital Division of Infectious Diseases and Harvard Medical School Department of Microbiology and Immunobiology, Boston, MA, United States; [3]Genomics Core Facility, European Molecular Biology Laboratory (EMBL), Heidelberg, Germany; [4]Department of Microbiology, The Ohio State University, Columbus, OH, United States; [5]Center for RNA Biology, The Ohio State University, Columbus, OH, United States

**Abstract** Adaptation to shifting temperatures is crucial for the survival of the bacterial pathogen *Vibrio cholerae*. Here, we show that colony rugosity, a biofilm-associated phenotype, is regulated by temperature in *V. cholerae* strains that naturally lack the master biofilm transcriptional regulator HapR. Using transposon-insertion mutagenesis, we found the *V. cholerae* ortholog of BipA, a conserved ribosome-associated GTPase, is critical for this temperature-dependent phenomenon. Proteomic analyses revealed that loss of BipA alters the synthesis of >300 proteins in *V. cholerae* at 22°C, increasing the production of biofilm-related proteins including the key transcriptional activators VpsR and VpsT, as well as proteins important for diverse cellular processes. At low temperatures, BipA protein levels increase and are required for optimal ribosome assembly in *V. cholerae*, suggesting that control of BipA abundance is a mechanism by which bacteria can remodel their proteomes. Our study reveals a remarkable new facet of *V. cholerae*'s complex biofilm regulatory network.

*For correspondence:
felipe.cava@umu.se

**Competing interests:** The authors declare that no competing interests exist.

## Introduction

A common strategy of bacteria for adaptation and survival to changing environmental conditions is the formation of biofilms, which are bacterial communities enclosed in an extracellular matrix. *Vibrio cholerae*, the causative agent of the severe human diarrhoeal disease cholera, forms biofilms both in biotic and abiotic surfaces in the aquatic environment that it inhabits (*Alam et al., 2007*; *Islam et al., 2007*; *Lutz et al., 2013*) and also in the intestine of the human host (*Faruque et al., 2006*; *Silva and Benitez, 2016*). Biofilm formation by *V. cholerae* provides protection against environmental insults, predators, and stress conditions and may also promote nutrient access (*Lutz et al., 2013*). There is some evidence indicating that biofilm formation is critical for intestinal colonization (*Silva and Benitez, 2016*), and it has been proposed that biofilm formation can enhance the infectivity of *V. cholerae* (*Tamayo et al., 2010*).

*V. cholerae*'s biofilm is primarily composed of *Vibrio* polysaccharide (VPS), matrix proteins (RbmA, RbmC, and Bap1), and extracellular DNA (*Berk et al., 2012*; *Fong et al., 2006*; *Fong and Yildiz, 2007*; *Fong et al., 2010*; *Reichhardt et al., 2015*; *Seper et al., 2011*; *Yildiz and Schoolnik, 1999*). The genes encoding the activities for the production of VPS, grouped in the *vpsI* and *vpsII*

clusters, and the genes encoding the RbmA and RbmC matrix proteins, located in the *rbm* cluster between *vpsI* and *vpsII* clusters, form the so-called *V. cholerae* biofilm-matrix cluster (*Fong et al., 2006*; *Fong and Yildiz, 2007*; *Fong et al., 2010*; *Yildiz and Schoolnik, 1999*).

Biofilm formation in *V. cholerae* is a highly regulated process, controlled by the transcriptional activators VpsR, VpsT, and AphA, the transcriptional repressors HapR and H-NS, small regulatory RNAs, alternative sigma factors (RpoS, RpoN, and RpoE), and small nucleotide signaling molecules (c-di-GMP, cAMP, and ppGpp). Specific environmental signals such as changes in salinity, osmolarity, nutrient availability, phosphate limitation, $Ca^{2+}$ levels, iron availability, and presence of polyamines (spermidine and norspermidine), indole or bile (*Conner et al., 2016*; *Teschler et al., 2015*) can also affect biofilm formation. Within this complex regulatory network, VpsR and VpsT, whose regulons extensively overlap (*Beyhan et al., 2007*), are the main transcriptional activators of the *vpsI* and *vpsII* clusters and the *rbmA*, *rbmC*, and *bap1* genes encoding the matrix proteins (*Zamorano-Sánchez et al., 2015*). HapR is the main repressor of biofilm formation and inhibits transcription of both the activators and the genes encoding the VPS and the matrix proteins (*Waters et al., 2008*). HapR also regulates other processes, such as virulence factor production, type VI secretion (*Kovacikova and Skorupski, 2002*; *Zheng et al., 2010*; *Zhu et al., 2002*), and intracellular c-di-GMP levels (*Hammer and Bassler, 2009*; *Waters et al., 2008*), which in turn indirectly affect biofilm formation. HapR expression is controlled by the quorum sensing (QS) cascade that responds to cell density (*Hammer and Bassler, 2009*; *Zhu and Mekalanos, 2003*), as well as by additional QS-dependent (*Lenz and Bassler, 2007*; *Liang et al., 2007*; *Shikuma et al., 2009*; *Tsou et al., 2011*) or QS-independent (*Liu et al., 2006*; *Yildiz et al., 2004*) regulators. There is considerable variation in the conservation of HapR function between and within the classical (pandemics 1–6), El Tor (pandemic 7), and variant El Tor (late pandemic 7) biotypes that stratify toxigenic *V. cholerae* (*Chowdhury et al., 2016*; *Hammer and Bassler, 2009*; *Joelsson et al., 2006*; *Katzianer et al., 2015*).

In both free-living and host-associated environments, *V. cholerae* must adapt to changing extracellular conditions. Specifically, *V. cholerae* experiences a wide range of temperatures, including seasonal and inter-annual temperature changes in the aquatic environment (ranging between 12 and 30°C; *Townsley et al., 2016*), and also upon infection of the human host (37°C). Once *V. cholerae* enters the human host, the temperature up-shift controls the expression of virulence factors (*Parsot and Mekalanos, 1990*; *Weber et al., 2014*). Adaptation to temperature is thus important not only for survival in the environment but also for the infection process and for subsequent transmission to a new host.

In this study, we identify *VC2744*, the *V. cholerae* ortholog of the translational GTPase BipA, as a critical determinant for repression of biofilm formation activity at low temperatures (i.e. <22°C). Loss of BipA leads to widespread shifts in the *V. cholerae* proteome at low temperatures, including increased production of the main biofilm transcriptional regulators VpsR and VpsT. Our data suggest that temperature could control BipA activity by altering its structural conformation and turnover. Finally, we show that the effects of BipA are only apparent in the absence of HapR, underscoring the intricacies of coordinating transcription and translation in response to temperature and cell density to govern biofilm development.

## Results

### Temperature governs colony morphology in *V. cholerae* HapR⁻ strains

Prolonged colony growth in *V. cholerae* can lead to the development of rugose colonies, which are tightly associated with biofilm formation (*Yildiz and Schoolnik, 1999*; *Yildiz and Visick, 2009*). We noticed that the rugose colony-forming *V. cholerae* El Tor clinical isolate co969 (*Figure 1A*) formed smooth colonies when cultured below 22°C (*Figure 1B*), suggesting a role for temperature in the regulation of this phenotype. Temperature-dependent rugosity was not dependent on culture time or culture density, as even at extended culture periods (up to 4 days) and comparable culture densities as 37°C cultures, c0969 *V. cholerae* did not form rugose colonies at 22°C (*Figure 1D*).

We first wondered whether the master biofilm regulator in *V. cholerae*, HapR, participated in this phenotype. *V. cholerae* co969 naturally lacks wild-type (WT) HapR due to a frameshift mutation that results in a truncated version of the protein (*Supplementary file 1* – Supplementary Figure 1C). We

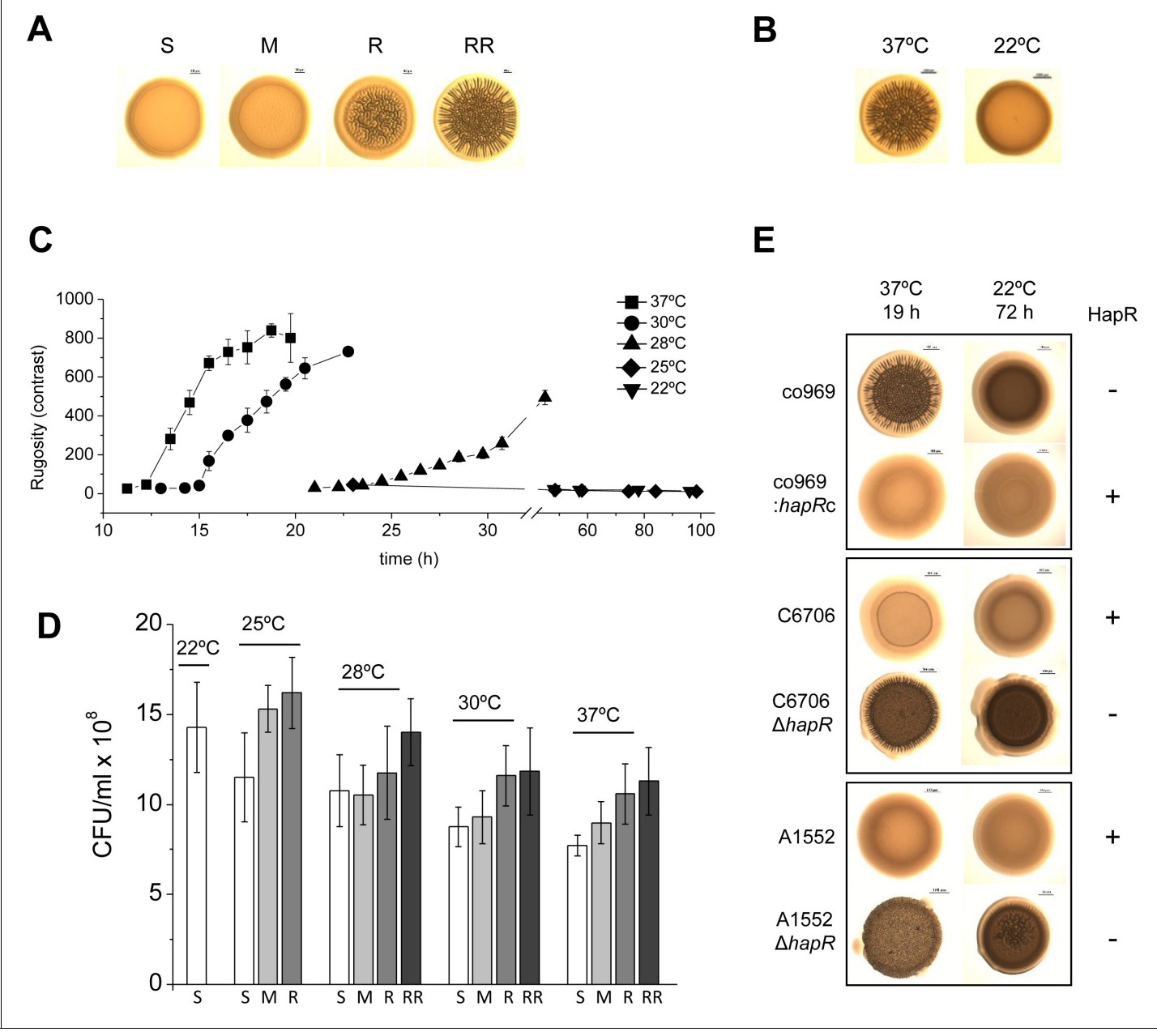

**Figure 1.** Development of *Vibrio cholerae* co969 colony rugosity is temperature- and HapR-dependent. (**A**) Representative images of different *V. cholerae* co969 colony rugosities at 37˚C: S (smooth), M (transition between smooth and rugose), R (rugose), and RR (very rugose). (**B**) Representative *V. cholerae* co969 colony morphologies at 37˚C and 22˚C, after incubation during 14 and 24 hr, respectively. (**C**) Development of *V. cholerae* co969 colony rugosity over time at different temperatures. Rugosity is represented as contrast calculated using ImageJ software (see Materials and methods section). Colonies grown at 22˚C remained smooth despite the incubation time. (**D**) Colony-forming units (CFUs) of collected colonies grown at different temperatures and different rugosity stages: S (smooth), M (transition between smooth and rugose), R (rugose), and RR (very rugose). Values are the average of at least three independent experiments with at least three biological replicates each. Error bars, standard deviation. For 22˚C, values are from smooth colonies after 48 hr incubation. (**E**) Colony morphology at 37˚C and 22˚C of co969, co969:*hapR*c (co969 strain carrying the active variant of *hapR* from C6706, *hapR*c), and C6706 and A1552 and their respective Δ*hapR*-mutant derivatives. The incubation times at different temperatures were previously optimized to result in colonies with a comparable number of CFU per colony.

observed that, similarly to co969, other *V. cholerae* WT strains with inactive HapR variants also exhibited temperature-dependent rugosity in both solid–air (i.e. colony morphology) and air–liquid interfaces (i.e. wrinkled pellicles; *Supplementary file 1* – Supplementary Figure 1). However, we did not observe this phenotype in HapR-sufficient *V. cholerae* strains, such as C6706 and A1552

(*Supplementary file 1* – Supplementary Figure 1), which were smooth at all temperatures. Remarkably, deletion of *hapR* in these strains led to temperature-dependent rugosity development comparable to the naturally *hapR*-deficient strains, and complementation of co969 with the WT *hapR* copy from C6706 (*hapRc*) negated its development of rugosity at high temperatures (*Figure 1E*).

## Biofilm genes are upregulated in colonies grown at 37°C vs. 22°C

As colony rugosity is a common marker of biofilm formation, we next measured the expression of specific structural biofilm components in co969 colonies grown at different temperatures using quantitative real-time polymerase chain reaction (qRT-PCR; *Figure 2A*). Both *vpsL*, one of the genes encoding the VPS, and *bap1*, encoding a biofilm matrix protein, were upregulated at 37°C vs. 22°C (correlating with the phenotypes in *Figure 1B*), suggesting that colony rugosity can be used as a readout in co969 to study temperature-dependent biofilm development regulation. *vpsL* and *bap1* expression did not differ between these two temperatures in the co969:*hapRc* background, consistent with the idea that the temperature-dependent program governing biofilm development in *V. cholerae* is enabled in the absence of HapR (*Figure 2A*).

To more broadly assess how the *V. cholerae* biofilm regulon was influenced by temperature in co969, we performed RNA-seq on rugose colonies grown at 37°C vs. smooth colonies grown at 22°C (37R vs. 22 Sb; *Figure 2*; *Figure 2—figure supplement 1*). We also performed control transcriptomic analyses of colonies collected at an earlier point, where these were still smooth (37S and 22S) to filter out temperature-independent or cell-density-driven gene expression changes (*Figure 2—figure supplement 1*). Consistent with our phenotypic observations, expression of the *vpsI* and *vpsII* gene clusters (e.g. *vpsL*), encoding the activities responsible for the production of the VPS, and the *rbmA*, *rbmC,* and *bap1* genes, encoding the biofilm matrix proteins, were upregulated (~10–20 fold) in 37R compared with 22Sb colonies (*Figure 2C*, *Figure 2—figure supplement 1*; *Supplementary files 3*, *4* and *5*). These results were validated by qRT-PCR (*Figure 2D*). Interestingly, most known biofilm transcriptional regulators (e.g. *vpsT, vpsR*, and *hapR*), as well as other genes involved in QS, type II secretion, and c-di-GMP signaling were not differentially expressed (*Figure 2C*; *Figure 2—figure supplement 1*; *Supplementary files 3, 4* and *5*). This suggested that temperature-dependent biofilm formation is controlled by either an unknown transcriptional regulator or a post-transcriptional mechanism.

## A genetic screen for determinants of temperature-dependent colony rugosity

We reasoned that our observations could stem from either activation (at 37°C) and/or repression (at 22°C) of gene expression. Since more is known about biofilm regulation at 37°C (*Teschler et al., 2015*) and mutagenesis at this temperature would produce a high number of false positives (i.e. all known structural biofilm proteins and temperature-independent activators), we decided to perform a loss-of-function screen for potential biofilm repressors at low temperatures. We performed random transposon mutagenesis on *V. cholerae* co969 and selected mutants that inappropriately formed rugose colonies at 22°C (*Figure 3A*). Out of 11,000 mutants screened, 459 colonies had a rugose phenotype at 22°C. Pooled sequencing of the 459 rugose colonies identified transposon insertions in 99 genes, corresponding to diverse cell functions such as flagellar motility, chemotaxis, c-di-GMP signaling, lipopolysaccharide biosynthesis, cell wall maintenance, phosphotransferase systems, regulatory functions, transport, and translation (*Figure 3B*; *Supplementary file 5*). To filter out potential temperature-independent repressors, we performed a similar screen for rugose colonies at 37°C using the temperature-insensitive, smooth *V. cholerae* strain C6706 (*Supplementary file 6*) and removed hits from this screen from our list. The most highly represented gene (frequently inserted gene [i.e. with the largest number of insertion reads]) after filtering out potential temperature-independent repressors in co969 rugose colonies at 22°C was *VC2744* (see *Supplementary file 5*). *VC2744* encodes a protein with 74% identity to *Salmonella enterica* serovar Typhimurium BipA, also referred to as TypA in other organisms (*Leipe et al., 2002*; *Margus et al., 2007*). Mutations in BipA have been associated with a cold-sensitive phenotype in *Escherichia coli* (*Pfennig and Flower, 2001*), but a clean deletion of *VC2744* in *V. cholerae* co969 did not influence growth or cell morphology at low or high temperatures (*Supplementary file 1* – Supplementary Figure 2). BipA has been implicated in biofilm formation in some bacteria (*Grant et al., 2003*; *Hiramatsu et al., 2016*;

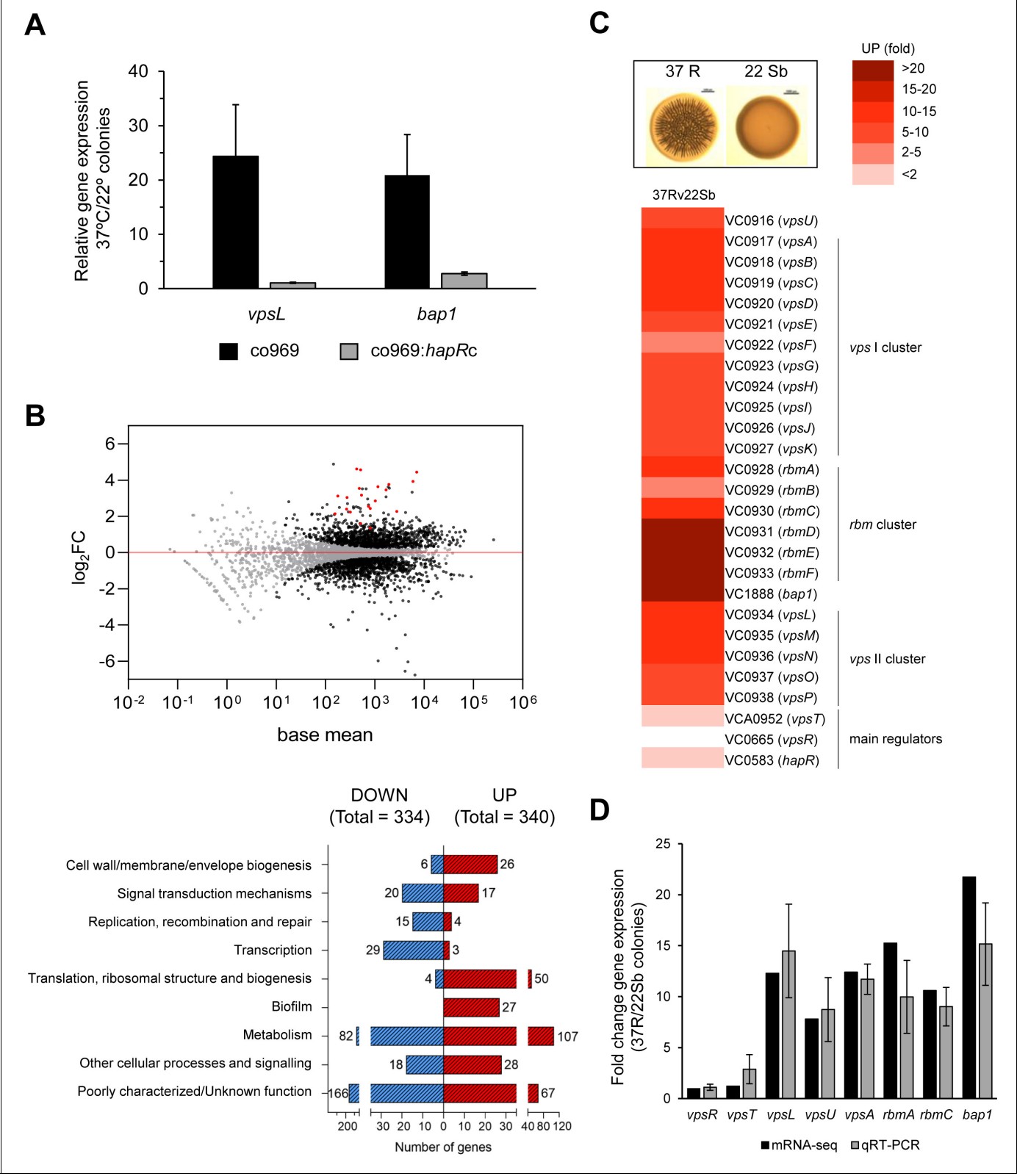

**Figure 2.** Biofilm structural components, but not regulators, are transcriptionally upregulated in rugose colonies grown at higher temperatures. Comparison of relative gene expression between *Vibrio cholerae* co969 colonies incubated at 37°C (rugose, 37R) and 22°C (smooth, 22 Sb) by mRNA-seq. Colonies had a similar number of colony-forming units per colony, as described under experimental procedures. (**A**) Relative expression of *vpsL* and *bap1* between colonies grown at 37°C vs. 22°C for co969 and co969:*hapR*c strains, determined by quantitative real-time polymerase chain reaction

*Figure 2 continued on next page*

*Figure 2 continued*

(qRT-PCR). Results are the average of three biological replicates, each replicate containing eight colonies. Error bars, standard deviation. Expression of *hfq* was used as a control. (B) Upper panel: MA-plot representing the log$_2$FC against mean expression for differentially expressed genes between 37R vs. 22 Sb colonies. Black dots, significantly differentially expressed genes; gray dots, not significantly differentially expressed genes; red dots, genes encoding biofilm structural components (*vpsI* and *vpsII* clusters, *rbmA*, *rbmC*, and *bap1*). Lower panel: number of differentially expressed genes (up- and downregulated) between 37R vs. 22 Sb colonies, grouped by Clusters of Orthologous Groups of proteins (COGs) categories. (C) Heat map showing the differential relative expression (fold change) between 37R and 22 Sb of genes belonging to the *vpsI*, *vpsII*, and *rbm* clusters and *bap1*, encoding the *Vibrio* polysaccharide (VPS) and biofilm matrix proteins (RbmA, RbmC, and Bap1), respectively, and main biofilm regulators VpsT, VpsR, and HapR. (D) Validation of differential gene expression obtained by mRNA-seq by qRT-PCR. The graph represents a comparison of the relative gene expression in both the mRNA-seq analysis and qRT-PCR experiments of biofilm activators (*vpsR* and *vpsT*), genes encoding VPS (*vpsL*, *vpsA*, and *vpsU*) and matrix proteins (*rbmA*, *rbmC*, and *bap1*) between 37R and 22 Sb colonies. Values are the average of three independent qRT-PCR experiments containing three biological replicates each, each replicate containing eight colonies pooled together and three technical triplicates of each biological replicate. Error bars, standard deviation. Expression of *gyrA* was used as a control.

The online version of this article includes the following figure supplement(s) for figure 2:

**Figure supplement 1.** Differential expression of biofilm-related genes between colonies grown at 37°C and 22°C, and collected at two different time points, by mRNA-seq.

*Neidig et al., 2013*; *Overhage et al., 2007*), but the mechanism underlying this phenotype has remained elusive.

## BipA represses rugose colony development at low temperature

Deletion of *VC2744* in co969 confirmed the rugose colony phenotype from our screen at 22°C (*Figure 3C*). Complementation of co969 Δ*VC2744* with either *V. cholerae VC2744* or *bipA* homologs from *E. coli* MG1655 K-12 or *Pseudomonas putida* KT2440 led to restoration of the WT smooth colony morphology phenotype at 22°C (*Figure 3D*). These results, together with the high degree of similarity between the proteins, strongly suggest that *VC2744* corresponds to the *bipA* ortholog in *V. cholerae*. Consistent with our previous data, deletion of *bipA* led to an increase in rugosity only in HapR⁻ (co969 or C6706 Δ*hapR*) but not HapR⁺ strains (C6706 WT or co969:*hapRc*), suggesting that the effect of BipA on colony morphology is epistatic to HapR in certain *V. cholerae* strains (*Supplementary file 1* – Supplementary Figure 3A and B).

A previous *V. cholerae* transcriptomic survey indicated that a mutant in the biofilm regulator *vqmA* increased *bipA* levels (*Liu et al., 2006*). However, deletion of *vqmA* in co969 did not phenocopy Δ*bipA* rugose colony morphology at low temperature, suggesting that VqmA does not affect the expression of *bipA* under these experimental conditions (*Supplementary file 1* – Supplementary Figure 3C). To place BipA in the hierarchy of biofilm regulators in *V. cholerae*, we also deleted this gene in the Δ*vpsR* and Δ*vpsT* backgrounds, both of which exhibit constitutively smooth colony morphologies. Both double mutants maintained smoothness (*Supplementary file 1* – Supplementary Figure 3C), suggesting that BipA acts upstream of VpsR and VpsT and further demonstrating rugose colony formation depends on the presence of biofilm regulatory elements. Collectively, these findings identify BipA as a regulator of temperature-dependent, biofilm-associated colony morphology in *V. cholerae*.

## BipA abundance is temperature-dependent

A key observation was that *bipA* overexpression at 37°C led to a very subtle reduction in rugosity compared to that at 22°C (*Figure 3D*), suggesting that BipA activity is higher or more consequential at lower temperatures. Remarkably, even though *bipA* transcript levels were largely unchanged between colonies grown at 37°C and 22°C (*Supplementary file 3* and *Figure 4A*), BipA protein levels were about 10 times higher at 22°C (*Figure 4B*). Temperature-dependent changes in BipA levels were observed both in the presence or absence of HapR (*Figure 4*), further suggesting that BipA regulation and its downstream effects are controlled by HapR.

We next performed circular dichroism (CD) to study whether BipA structure could be regulated by temperature. Interestingly, purified BipA pre-incubated at different temperatures (37°C, 22°C and 15°C) exhibited slightly different CD curves (*Supplementary file 1* – Supplementary Figure 4). Furthermore, incubation of BipA at 37°C followed by subsequent incubation at 22°C showed the same

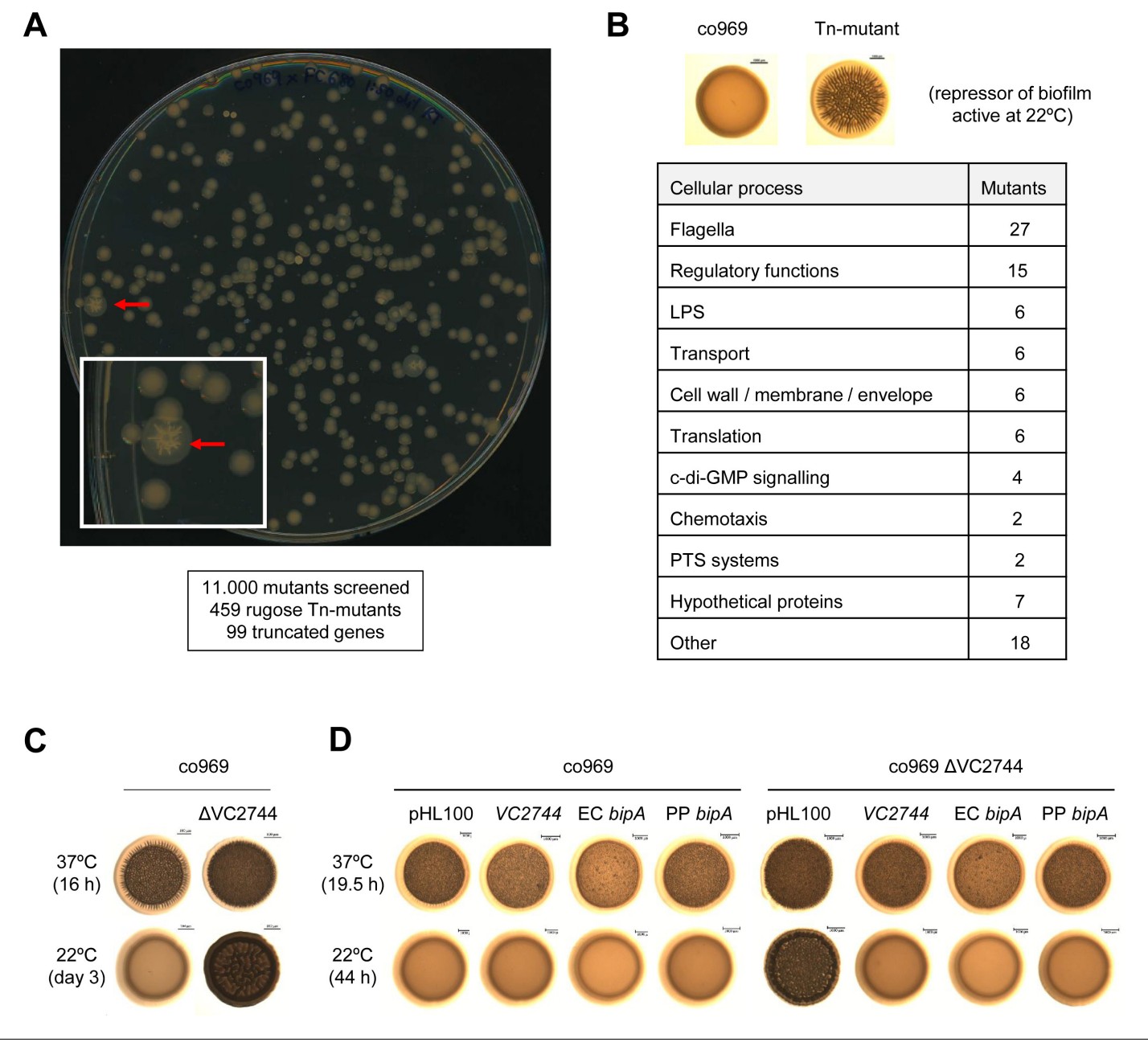

**Figure 3.** Transposon mutagenesis identifies VC2744 as a regulator of *Vibrio cholerae* colony morphology at 22°C. (**A**) Representative image of an agar plate used for the selection of *V. cholerae* co969 transposon mutants with a rugose colony phenotype (pointed with red arrows) at 22°C. The number of screened transposon mutants, rugose colonies selected, and final number of genes with insertions leading to truncations is indicated below. (**B**) Table of transposon screen hits sorted by functional annotation. (**C** and **D**). Effect of VC2744 on co969 colony morphology. (**C**) Deletion of VC2744 in co969 results in a rugose colony phenotype at 22°C. (**D**) Colony morphology at 37°C and 22°C of co969 and co969 Δ*VC2744* carrying either pHL100-*bipA*, for overexpression of *V. cholerae* co969 *VC2744* from the isopropyl-β-d-thiogalactosidase-inducible P*lac* promoter, pHL100-EC*bipA*; or pHL100-PP*bipA*, for overexpression of the *bipA* variants of *Escherichia coli* MG1655 K-12 or *Pseudomonas putida* KT2440, respectively, or the empty plasmid (pHL100). Overexpression of the different BipA variants restored the smooth colony phenotype at 22°C, while it only resulted in slightly decreased rugosity at 37°C.

pattern as for the protein incubated only at 37°C, suggesting that temperature-dependent protein folding changes in BipA that are likely irreversible.

Reduced BipA stability and protein levels at 37°C suggested a potential proteolytic control mechanism. Therefore, we aimed to identify the BipA-targeting protease(s). Transposon mutagenesis

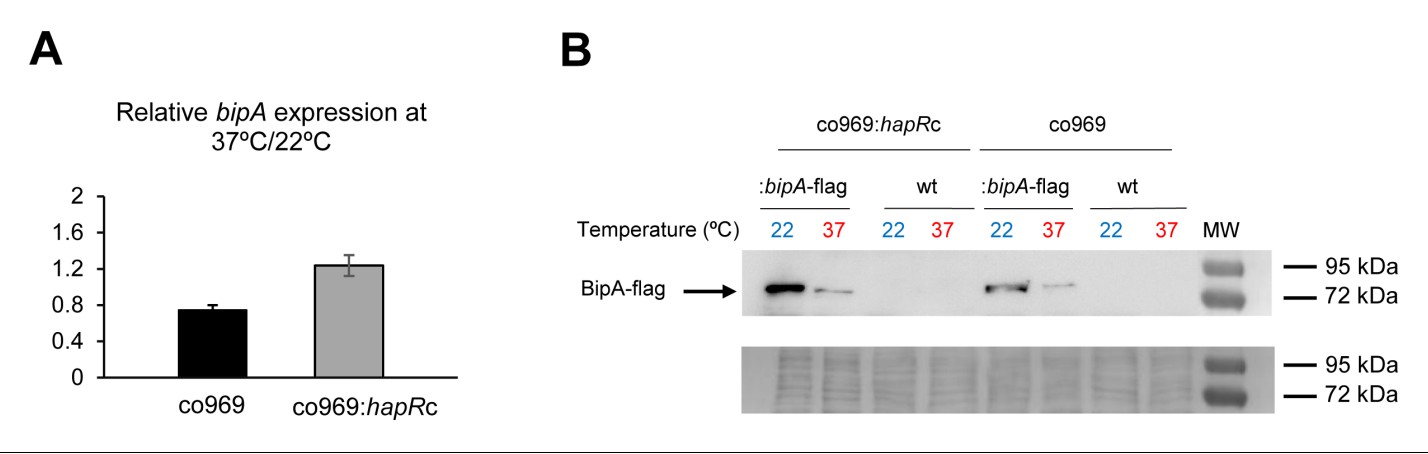

**Figure 4.** BipA protein levels are elevated at 22°C. (**A**) Relative *bipA* transcript levels in *Vibrio cholerae* co969:*bipA*-flag or co969:*hapRc*:*bipA*-flag strains, carrying a chromosomal *bipA*-flag fusion, between rugose colonies grown at 37°C (37R) and smooth colonies grown at 22°C (22 Sb), determined by quantitative real-time polymerase chain reaction. Data are the average of three biological replicates, each containing eight colonies pooled together. Error bars, standard deviation. (**B**) Representative Western blot showing BipA-flag (68.18 KDa) protein levels of co969:*bipA*-flag or co969: *hapRc*:*bipA*-flag rugose colonies grown at 37°C (37R) and smooth colonies grown at 22°C (22 Sb). BipA-flag bands migrated close to the 72 KDa band of the protein ladder. co969 and co969:*hapRc* strains were used as negative controls. The Coomassie staining of the membrane, used as a loading control, is shown.

screening for aberrantly smooth co969 colonies at 37°C identified the protease LonA, which has previously been reported to participate in biofilm regulation, as a potential repressor of BipA activity (***Supplementary file 1*** – Supplementary Figure 5; ***Supplementary file 7; Rogers et al., 2016***). Deletion of VC1920 (*lonA*) in co969 led to the formation of smooth colonies at both 37°C and 22°C (***Supplementary file 1*** – Supplementary Figure 5B), and overexpression of *lonA* resulted in increased rugosity at both temperatures (***Supplementary file 1*** – Supplementary Figure 5C). However, BipA protein levels were unchanged in the Δ*lonA* background (***Supplementary file 1*** – Supplementary Figure 5D), suggesting that the effect of LonA in biofilm formation does not involve BipA. Accordingly, deletion of *lonA* in the co969 Δ*bipA* background also resulted in smooth colonies at all temperatures, and overexpression of *lonA* in co969 Δ*bipA* further increased rugosity (***Supplementary file 1*** – Supplementary Figure 5B and C). These data show that LonA regulates biofilm-associated colony morphology in *V. cholerae*, but likely in a BipA-independent manner.

## BipA contributes to 50S subunit assembly in *V. cholerae*

In *E. coli*, BipA is thought to act as a ribosome assembly factor, facilitating 50S subunit biogenesis at suboptimal temperatures (***Choi et al., 2019***; ***Choudhury and Flower, 2015***; ***Gibbs et al., 2020***; ***Krishnan and Flower, 2008***). We reasoned that the role of BipA in *V. cholerae* colony morphology development might be related to ribosome biogenesis and/or homeostasis. To test this, we prepared lysates from control and Δ*bipA* cells grown at different temperatures and subjected them to sucrose gradient sedimentation analysis. We found that cells grown at 22°C and lacking BipA had increased proportions of subunit particles (***Figure 5***). The 50S peak also exhibited a shoulder of slower-migrating particles. At 37°C, no differences were observed between control and mutant strain. These data are consistent with a modest 50S assembly defect at low temperature, similar to that reported for *E. coli* (***Gibbs et al., 2020***).

## BipA influences translation at lower temperatures

To investigate the role of BipA in control of biofilm component genes and colony morphology, we performed global proteomic analyses of co969 WT and Δ*bipA* grown at either 37°C or 22°C followed by four comparative analyses: (i) WT vs. Δ*bipA* at 37°C, (ii) WT vs. Δ*bipA* at 22°C colonies, (iii) WT 37 vs. 22°C colonies, and (iv) Δ*bipA* 37 vs. 22°C colonies. The proteomic analyses identified 1639 (43%) of 3783 known *V. cholerae* proteins, out of which 695 were significantly differentially abundant (***Supplementary file 8***).

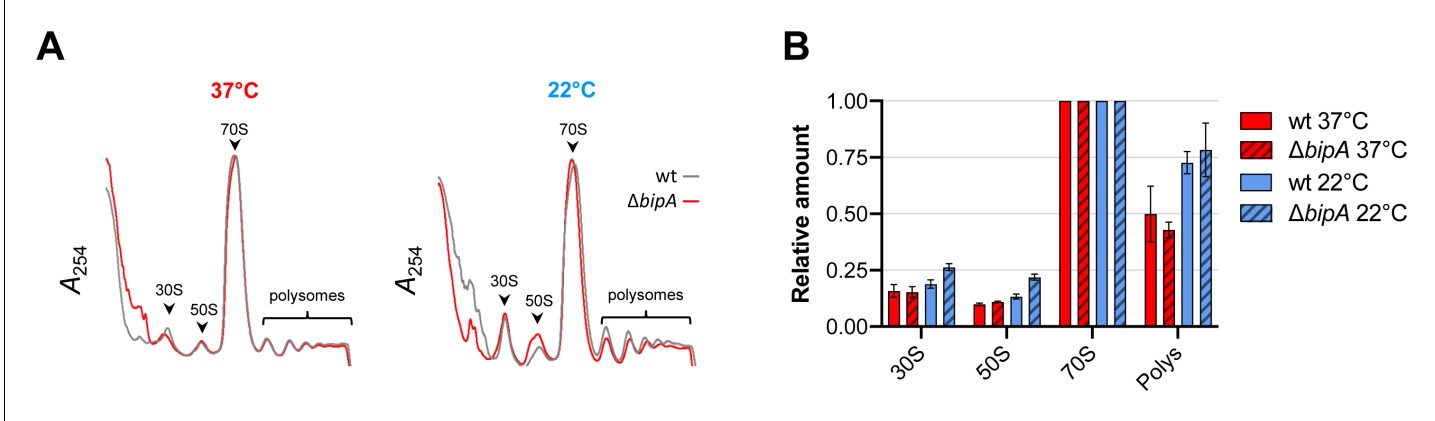

**Figure 5.** Ribosome assembly analyses in the wild-type and the *bipA* mutant. (**A**) Representative traces of sucrose gradient sedimentation experiments, involving cells grown at 37°C or 22°C, as indicated. Absorbance at 254 nm ($A_{254}$) is shown from the top to the bottom of the gradient (left to right), and peaks corresponding to 30S, 50S, 70S, and polysomes (multiple ribosomes per mRNA) are indicated. (**B**) Levels of 30S particles (30S), 50S particles (50S), and polysomes (Polys), normalized with respect to 70S monosomes (70S), from various cells as indicated.

Proteomic differences between co969 WT and Δ*bipA* were greater at 22°C than at 37°C, suggesting that BipA is more active or influential at lower temperatures (*Figure 6A and B*). Relative to the WT strain, 250 proteins were more abundant in Δ*bipA* cells at 22°C, and 52 proteins were less abundant (*Figure 6C*). The most represented differentially produced proteins were related to external cellular components and biological processes involved in localization and transport (*Figure 6C*). We identified a number of biofilm components among the upregulated proteins, confirming that BipA acts to inhibit biofilm formation-associated processes at low temperatures (i.e. 22°C). Consistent with this finding, the co969 Δ*bipA* mutant exhibited 10–20% reduced motility compared to the WT strain, with motility slightly more reduced at 22°C than at 37°C (*Supplementary file 1* – Supplementary Figure 6A). We observed similar phenotypes in the co969:*hapR*c background (*Supplementary file 1* – Supplementary Figure 6B), indicating that the effect of BipA on motility is independent of the presence of HapR.

### Translation of biofilm genes is reduced by the presence of BipA at low temperature

To validate the effects of BipA on the production of biofilm-associated genes in the proteomic analysis (*Figure 6A–C*; *Supplementary file 8*), we carried out β-galactosidase assays with co969 Δ*bipA* and co969 strains carrying translational reporter fusions to several biofilm components or regulatory genes (*vpsR, vpsT, vpsL, vpsU,* and *bap1*). Additionally, we included translational reporter fusions to housekeeping genes, *gyrA* and *hfq*, and the cell wall biosynthesis gene, *mrcA*, as controls. The results demonstrated increased translation of all biofilm-related proteins in the co969 Δ*bipA* mutant vs. WT, especially at 22°C (*Figure 6D*). Critically, no substantial changes were observed for *gyrA*, *hfq*, and *mrcA*, indicating that production of these proteins is largely independent of BipA.

Akin to the biofilm structural genes, translation of *vpsR* and *vpsT* was higher in the co969 Δ*bipA* background, especially at 22°C (*Figure 6D*). However, compared to the biofilm structural genes, the transcriptional levels of these two major biofilm regulators were mostly unchanged at different temperatures and between Δ*bipA* and WT (*Figure 6—figure supplement 1*; *Supplementary files 3*, *4* and *5*). Collectively, these results suggest that BipA influences the production of many proteins at 22°C, including key biofilm regulators, potentially explaining the importance of BipA in temperature-dependent formation of biofilm-associated colony morphology.

## Discussion

The molecular mechanisms that underlie biofilm development in *V. cholerae* are complex and rely on many different inputs and pathways that converge on the regulation of the core biofilm regulon.

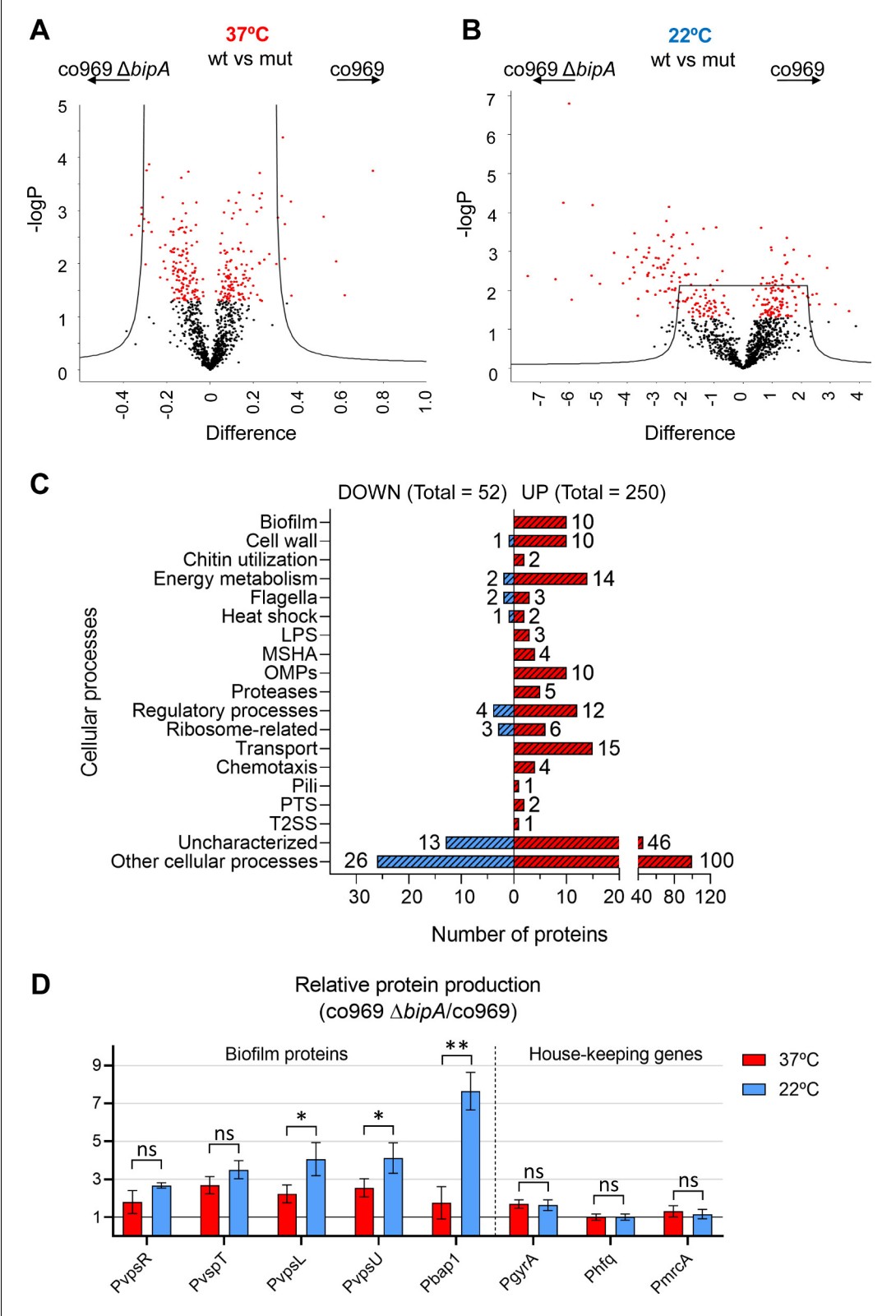

**Figure 6.** Global proteomic analysis of *Vibrio . cholerae* co969 wild-type (WT) vs. Δ*bipA* colonies grown at 37°C and 22°C. (A and B) Volcano plots representing the log t-test p-value against the t-test difference for a comparison of protein levels between co969 WT vs. Δ*bipA* colonies grown at either 37°C (A) or 22°C (B). Black dots represent proteins belonging to the pool of non-differentially produced proteins between the WT and the Δ*bipA* strain; red dots represent proteins that are significantly different between both strains at the given condition. Higher numbers of t-test difference and log

*Figure 6 continued on next page*

*Figure 6 continued*

t-test p-value indicate more differentially produced proteins. (**C**) Differentially produced proteins between co969 WT vs. Δ*bipA* colonies at 22℃ grouped by function. (**D**) Relative translation of biofilm-related and control proteins between co969 Δ*bipA* and co969 strains, as measured by translational *lacZ* fusions in colonies grown at 37℃ (red) or at 22℃ (blue). Relative Miller activity in β-galactosidase assays was calculated from the average of two independent experiments containing three biological replicates each, each replicate containing four colonies. Error bars, standard deviation. t-test: *p<0.05; **p<0.01; ns: not significant.

The online version of this article includes the following figure supplement(s) for figure 6:

**Figure supplement 1.** Effect of BipA on expression of biofilm-related genes.

Previous studies have described the existence of a complex interplay of regulatory mechanisms that ultimately result in the transcriptional control of the biofilm genes. Here, complementary genetic and proteomic approaches led us to discover that BipA is critical for temperature-dependent changes in production of biofilm components and altered colony morphology in *V. cholerae* HapR⁻ strains. Loss of BipA alters the levels of >300 proteins in *V. cholerae* grown at suboptimal temperature, increasing the relative levels of 250 proteins, including virtually all known biofilm-related proteins in this pathogen. Thus, BipA not only impacts the expression of the biofilm formation program but also likely shapes other aspects of pathogen physiology at low temperatures.

BipA is a broadly conserved translational GTPase whose precise cellular function has remained elusive (*Gibbs and Fredrick, 2018*). A growing body of evidence indicates that BipA facilitates 50S subunit biogenesis at suboptimal temperature. *E. coli* K12 cells lacking BipA exhibit cold sensitivity and accumulate immature 50S particles when grown at suboptimal temperature (*Choudhury and Flower, 2015*; *Gibbs et al., 2020*; *Krishnan and Flower, 2008*). Recent evidence shows that mature free 30S subunits accumulate in the absence of BipA, presumably due to a shortage of available 50S partners (*Gibbs et al., 2020*). In line with these *E. coli* studies, our sucrose gradient sedimentation analyses indicate that BipA is important for 50S subunit assembly in *V. cholerae*, specifically at low temperatures. In the mutant strain, larger subunit peaks (which include immature particles) are observed, as is a <50S shoulder, characteristics of an assembly defect. Notably, compared to that of *E. coli*, this 50S assembly defect in *V. cholerae* is more subtle, which may explain why no obvious growth defect is observed at 22℃ (*Supplementary file 1* – Supplementary Figure 2).

How does loss of BipA cause de-repression of biofilm formation genes at 22℃? One possibility is that the defect in 50S assembly alters the free subunit concentrations in the cell, which has variable consequences on translation depending on the particular mRNA (*Mills and Green, 2017*). Initiation of translation entails two major steps − formation of the 30S initiation complex and docking of the 50S subunit. An increase in free mature 30S subunits and/or decrease in free mature 50S subunits would be expected to enhance translation of certain mRNAs, reduce translation of other mRNAs, and have no impact on translation of still other mRNAs. We speculate that such perturbed translation in the absence of BipA results in increased production of one or more key regulatory proteins, like VpsR and/or VpsT, leading to transcriptional activation of the entire biofilm regulon (*Figure 7*). Consistent with this general model, *E. coli* cells lacking LepA, a related translational GTPase involved in 30S biogenesis (*Gibbs et al., 2017*), show numerous changes in protein synthesis that depend on specific mRNA features (*Balakrishnan et al., 2014*).

Our observation that HapR⁺ strains do not undergo temperature-dependent shifts in biofilm conflicts with results previously reported for *V. cholerae* A1552 and *Vibrio salmonicida*, which both encode functional HapR homologs. In *V. cholerae* A1552, increased c-di-GMP production at low temperatures by specific diguanylate cyclases (DGCs) and the cold-shock gene *cspV* lead to higher biofilm formation (*Townsley et al., 2016*; *Townsley and Yildiz, 2015*). However, it is important to note that we did not perform identical assays to the previous groups. For instance, biofilm and colony morphology might not be strictly equivalent phenomena or subtle changes in experimental conditions might also influence the impact that c-di-GMP production has on biofilm-associated phenotypes in this strain. Furthermore, the six DGCs that were induced at low temperature in *V. cholerae* A1552, and account for the increased biofilm production at low temperature, did not appear to be differentially expressed in our RNA-seq studies of co969. Since it is known that *V. cholerae* El Tor and classical biotypes (HapR⁺ and HapR−, respectively) modulate c-di-GMP levels using different pathways (*Hammer and Bassler, 2009*), which in turn results in differential regulation of

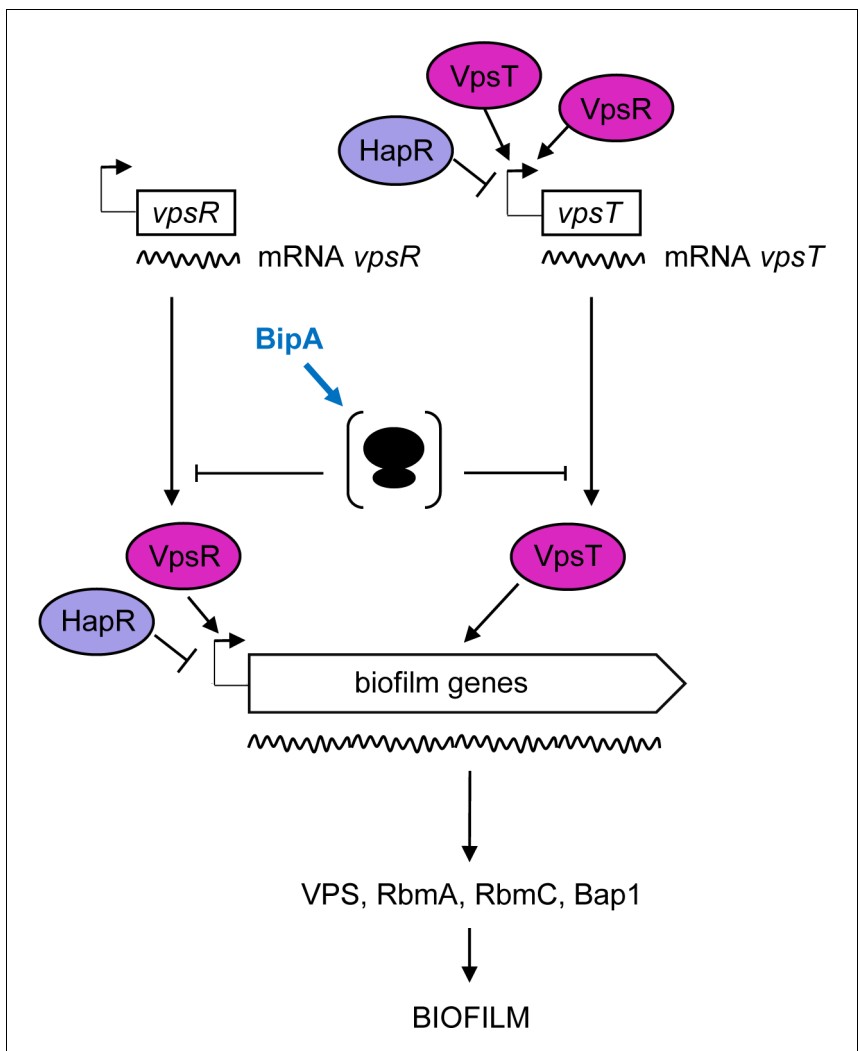

**Figure 7.** Proposed model for BipA within the *Vibrio cholerae* biofilm regulatory cascade. Schematic showing the proposed model for the interplay between the main biofilm transcriptional regulators, VpsR, VpsT, and HapR, and the translational repressor, BipA. In HapR[−] strains, transcription of *vpsR* and *vpsT* and the biofilm genes (*vpsI* and *vpsII* clusters encoding the *Vibrio* polysaccharide [VPS], and *rbmA, rmbC,* and *bap1* encoding the matrix proteins) leads to biofilm formation at 37°C but not at 22°C, where high levels of BipA inhibit translation of the mRNAs of the biofilm activators and/or structural genes. In HapR[+] strains, transcription of both the biofilm activators and the biofilm genes is negatively regulated by HapR, and no biofilm is produced (i.e. a smooth colony forms). At 22°C, BipA constitutes an additional layer of control by ensuring that, even if residual levels of biofilm-associated transcripts are produced, their translation would be inhibited.

biofilm formation, it seems likely that the presence or absence of HapR could account for the observed differences in the temperature-dependent regulation between these *V. cholerae* strains. These results further underscore the diversity of biofilm regulatory processes in pathogenic *V. cholerae,* especially in strains where the master regulator HapR is absent.

The data presented in this work show that HapR neither regulates BipA production (*Figure 4A and B*) nor influences the effect of Δ*bipA* on cell motility (*Supplementary file 1* – Supplementary Figure 6). Although HapR or BipA alone (HapR[+] strains and high temperature or HapR[−] strains at low temperature, respectively) are sufficient to prevent rugose colony morphology (*Supplementary file 1* – Supplementary Figure 3A and B), these proteins both contribute to the repression of biofilm gene expression via transcriptional and translational control. Moreover, our proteomics data showed that BipA, like HapR, impacts many processes including virulence,

competence, QS, and protein secretion. We propose that biofilm formation represents just one of several cellular programs whose regulation involves combined transcriptional and translational control.

Since BipA is highly conserved in prokaryotes (*Gibbs and Fredrick, 2018*; *Leipe et al., 2002*; *Margus et al., 2007*), it is possible that temperature-dependent control of biofilm gene expression by BipA observed in *V. cholerae* is conserved in other bacteria. In agreement with this hypothesis, deletion of *bipA* has been associated with increased biofilm levels in *Pseudomonas aeruginosa* PAO1 (*Neidig et al., 2013*) and *Bordetella holmesii* (*Hiramatsu et al., 2016*). BipA from *E. coli* and *P. aeruginosa* cross-complemented *V. cholerae* Δ*bipA* temperature-dependent phenotypes (*Figure 3D*), further suggesting that BipA plays a similar role in protein synthesis at suboptimal temperature across various Proteobacteria.

In both *E. coli* and *V. cholerae*, BipA levels are regulated by temperature. However, the basis of regulation appears to be transcriptional in the case of *E. coli* (*Choi and Hwang, 2018*) and post-transcriptional in the case of *V. cholerae*. Steady-state levels of BipA increase substantially in *V. cholerae* at reduced temperature, without an increase in transcript levels (*Figure 4*). CD experiments suggest that BipA adopts a conformation at 37°C that makes the protein more susceptible to degradation. These data raise the possibility that temperature-dependent control of BipA is governed by the intrinsic stability of the protein in the cell. Since bacteria thrive within a wide range of temperatures, it would be interesting to identify the amino acid residues in the sequence of BipA that dictate its capacity to change conformation, stability, and half-life in response to temperature fluctuation. Differences in the sequence of BipA may also explain distinct regulatory outcomes between species. For example, BipA is crucial for resistance to the antimicrobial peptide P2 in *E. coli* and *Salmonella* spp. (*Barker et al., 2000*; *Sy et al., 1995*), but not in *V. cholerae* (*Mathur and Waldor, 2004*).

Collectively, defining the cellular processes and regulatory networks that lay under the control of BipA will shed light on how *V. cholerae* coordinates multiple behaviors in response to temperature changes. Our study supports and provides an explanation for recent findings suggesting a role of BipA in promoting biofilm dispersion in *V. cholerae* (*Bridges et al., 2020*). Future research will address the effect of temperature- and BipA-dependent regulation of bacterial physiology during host-environment transitions and the associated potential consequences in cholera transmission and outbreaks.

## Materials and methods

### Bacterial strains and culture conditions

All *V. cholerae* and *E. coli* bacterial strains used in this study are listed in *Supplementary file 2* – Supplementary Table 1. All strains were cultured in Luria–Bertani (LB)/Lennox medium (containing 5 g/l NaCl for *E. coli* and 10 g/l NaCl for *V. cholerae*) at 37°C or at the otherwise stated temperature. Cultures were supplemented with carbenicillin (Cb, 100 µg/ml), kanamycin (Km, 50 µg/ml), or streptomycin (Sm, 200 µg/ml for *V. cholerae*) when appropriate for strain or plasmid selection.

*V. cholerae* clean deletion mutant strains were generated by recombination as follows. The corresponding mobilizable R6K-based suicide pCVD442-derivative plasmids, carrying an insertion with the upstream and downstream regions of the gene to be deleted, were conjugated from *E. coli* SM10 *λ* pir into the appropriate *V. cholerae* strain. First, single-site recombinants were selected by plating on LB (10 g/l of NaCl) plates in the presence of Sm and Cb. Then, LB plates containing 10% (w/v) sucrose were used for the selection of double recombinants, which were later verified by screening for loss of the suicide plasmid by testing sensitivity to Cb and by PCR with the appropriate primers (*Supplementary file 2* – Supplementary Table 3).

### Plasmid constructions

Plasmids (*Supplementary file 2* – Supplementary Table 2) were constructed by standard DNA cloning techniques. The fidelity of the DNA regions generated by PCR amplification was confirmed by DNA sequencing.

β-galactosidase translational reporter plasmids are based on pCB192N plasmid (*Schneider et al., 2012*). Basically, 300–600 bp putative promoter regions were PCR-amplified using the

corresponding primers (*Supplementary file 2* – Supplementary Table 3) and *V. cholerae* co969 DNA as template, and cloned between HindIII or BglII and EcoRI of pCB192N.

Plasmids used to make clean deletion mutants were based on pCVD442 (*Donnenberg and Kaper, 1991*) and were constructed in a step-wise manner. First, the chromosomal regions upstream and downstream of the genes to be deleted were PCR amplified using the corresponding primer pairs P1–P2 and P3–P4 and *V. cholerae* co969 DNA as template. The resulting PCR products were mixed and used as DNA template for a second round of PCR with primers P1 and P4. The amplified PCR products were then digested and cloned into the specific restriction sites of pCVD442: XbaI for pCVD442 Δ*hapR*, pCVD442 Δ*vpsR*, pCVD442 Δ*vqmA*, and pCVD442 Δ*lonA*; SalI for pCVD442 Δ*vpsT*; and SacI for pCVD442 Δ*bipA*.

Plasmids used for complementation and/or overexpression were based on pHL100 (*Cava et al., 2011*). The corresponding genes were PCR amplified with the primers listed in *Supplementary file 2* – Supplementary Table 3, digested with the appropriate restriction enzymes and cloned into the specific sites of pHL100 (*Supplementary file 2* – Supplementary Table 2).

For construction of plasmid pET22b-*bipA*-His used for purification of BipA-His, *bipA* was amplified as a NdeI-HindIII fragment carrying a His tag before the codon stop (primers in *Supplementary file 2* – Supplementary Table 3) and inserted into the same sites of pET22b(+) (Novagen).

## Colony morphology assays

Two microlitre drops of *V. cholerae* cultures grown over day (o/d cultures), started from a 1:100 dilution of a culture grown over night (o/n culture), were dispensed on LB agar plates (10 g/l NaCl) and incubated for the indicated times at the indicated temperature (37, 30, 28, 25, or 22°C). The morphology of the colonies was recorded using a Nikon SMZ1500 stereomicroscope.

Colony rugosity was monitored by measuring contrast using the Gray Level Correlation Matrix (GLCM) Texture Analyzer plugin of ImageJ (*Schneider et al., 2012*). The contrast values report on the heterogeneity/rugosity of the colonies by measuring differences between two adjacent pixels in the colonies. Higher contrast values correlate to higher rugosity.

## Pellicle biofilm assays

Five microlitre drops of *V. cholerae* o/d cultures, started from a 1:100 dilution of an o/n culture, were dispensed into 12 well plates, each well containing 5 ml of LB (10 g/l NaCl), and incubated the appropriate time at the mentioned temperature (37, 30, 28, 25, or 22°C). Images of pellicle biofilms were recorded using a Nikon SMZ1500 stereomicroscope.

## Growth and viability assays

For growth curves, stationary cultures of both strains were normalized to an $OD_{600}$ of 0.4 and used for inoculating 96-well plates containing 180 μL of fresh LB medium (10 g/L NaCl). $OD_{600}$ measurements were carried at 37, 30, or 22°C with shaking using an Eon Biotek microplate spectrophotometer. Nine replicas for each strain were grown and the assay was performed twice.

For viability assays, stationary cultures of *V. cholerae* co969 WT and Δ*bipA* mutant were normalized to an optical density of 600 nm ($OD_{600}$) of 0.5. Next, 10-fold serial dilutions were prepared and 10 μL spotted into LB agar plates. Plates were incubated at either 37°C or 22°C and then imaged using an LAS-3000 Imaging System (Fuji). At least three replicates per strain and condition were carried out.

## Motility assays

Two microliter drops of *V. cholerae* o/n cultures were dispensed on 0.3% LB agar plates (10 g/l NaCl) and incubated for the indicated time at 37 or 22°C. For *V. cholerae* strains carrying pHL100 derivatives, 0.3% LB agar plates (10 g/l NaCl) + Km (50 μg/ml) + IPTG (isopropyl-β-d-thiogalactosidase, 1 mM) were used. Motility was determined by measuring the radius using ImageJ (*Schneider et al., 2012*).

## Total RNA isolation from *V. cholerae* colonies

Total RNA from *V. cholerae* co969 colonies grown at 37°C or 22°C was isolated as follows: eight colonies grown under the desired conditions were collected by scraping, pooled together in an Eppendorf tube for RNA extraction and stored at −20°C. For comparative analysis, the $OD_{600}$ of the resuspended colonies, as well as their CFU per colony and total protein content were similar for the colonies grown at 37°C and 22°C at the selected sample collection time points: time 1 (24 hr for smooth colonies at 22°C [22S] and 12 hr for smooth colonies at 37°C [37S]) and time 2 (24 hr for smooth colonies at 22°C [22 Sb] and 18 hr for rugose colonies at 37°C [37R]). Pellets were resuspended in 400 µl of solution containing 10% glucose, 12.5 mM Tris pH 7.6 and 5 mM Ethylenediaminetetraacetic acid (EDTA) and, after adding 60 µl 0.5 M EDTA, cells were disrupted on a bead beater (4°C, maximum speed for 1 min 15 s) in the presence of 0.5 ml of acid phenol. After centrifugation, 1 ml Trizol (Ambion) was added to the supernatant, incubated at room temperature for 10 min, 100 µl chloroform:IAA were added, mixed by vortex for 10 s, and centrifuged at 14,000 rpm 4°C for 15 min. After two more chloroform:IAA extractions, the RNA was precipitated by mixing the aqueous phase with 0.7 volumes isopropanol, incubated for 30 min at −20°C, centrifuged at 14,000 rpm 4°C for 30 min, and washed with 70% ethanol. RNA pellets were then resuspended in 200 µl $H_2O$ and subjected to two consecutive DNaseI (Roche) treatments for 1 hr at 37°C, in the presence of RNase inhibitor. After a phenol:chloroform:IAA and a chloroform:IAA extraction, the RNA was precipitated by mixing the aqueous phase with three volumes of 95% ethanol and 1/10 volumes of 3M sodium acetate pH 4.6, incubated for 30 min at −20°C, centrifuged at 14,000 rpm 4°C for 30 min, washed with 70% ethanol, and finally resuspended in 100 µl $H_2O$. The integrity of the RNA was checked on a 1.2% agarose gel and with RNA6000 Nano Assay using the Agilent 2100 Bioanalyzer (Agilent Technologies), and RNA samples were quantified using a Qubit 2.0 Fluorometer (Life Technologies).

## RNA sequencing

To enrich mRNA, ribosomal RNA (rRNA) was removed from the samples using the Ribo-Zero rRNA removal kit (Illumina). cDNA preparation and sequencing reactions were conducted in the GeneCore Facility of the European Laboratory for Molecular Biology (EMBL, Heidelberg). Construction of mRNAseq libraries was conducted using the TruSeq Stranded Total RNA Library Prep Kit (Illumina) following the manufacturer's recommendations. The samples were clustered on a flow cell and 50 cycle paired-end sequencing was performed on an Illumina HiSeq 2000 (Illumina).

For the data analysis, raw sequencing data generated from Illumina HiSeq2000 was converted into fastq files and de-multiplexed. Data analysis was performed using R version 3.6.2. Quality of the fastq files was checked with FastQC report version 0.11.4 and sequence reads were trimmed with Trimommatic version 0.36 (*Bolger et al., 2014*) to remove adapters, primers and reads with low quality. The resulting sequence reads were aligned to the reference genome for *V. cholerae* N16961 using Bowtie version 1.1.1 (*Langmead and Salzberg, 2012*) and the obtained .sam files were converted into .bam files with Samtools version 1.4 (*Li, 2011*; *Li et al., 2009*). For each gene, read counts were calculated using in-house Perl scripts. Differential gene expression analysis was performed with DESeq2/Bioconductor version 1.28.29 (*Love et al., 2014*). Principal component analysis was performed to examine sample separation between the two groups.

The data for this study have been deposited in the European Nucleotide Archive (ENA) at EMBL-EBI under accession number PRJEB42488 (https://www.ebi.ac.uk/ena/browser/view/PRJEB42488).

## Quantitative real-time PCR

Total RNA was isolated from *V. cholera* colonies as described above. For cDNA synthesis, 4 µg of total RNA was incubated for 5 min at 65°C, to avoid formation of secondary structures, in the presence of dNTPs and random hexamer primers. Then, 200 U of Maxima RT (Thermo Scientific) and RNase inhibitor were added to the mix and further incubated 10 min at 25°C, 30 min at 50°C, and finally 5 min at 85°C, for heat inactivation of the enzyme. For RT control reactions, to verify the absence of contaminant genomic DNA in the RNA samples, $H_2O$ was added instead of Maxima RT. Once the reactions were completed, ~5 U RNase A (Thermo Scientific) and 2.5 U RNase H (Thermo Scientific) were added to the reaction mixtures and incubated for 30 min at 37°C to remove the remaining DNA. The synthesized cDNA was purified using a QIAquick PCR purification kit (Qiagen),

and its concentration was determined spectrophotometrically in a NanoDrop Lite Spectrophotometer (Thermo Scientific). qRT-PCR was performed using an iQ 5 Multicolor Real-Time PCR Detection System (Bio-Rad) with qPCRBIO SyGreen Mix fluorescein (PCR BIOSYSTEMS). Master mixes were prepared as recommended by the manufacturer, with qRT-PCR primers listed in *Supplementary file 2* – Supplementary Table 3. For normalization, either *gyrA* or *hfq* was used as internal standards.

## Transposon mutagenesis of *V. cholerae*

Transposon mutagenesis was performed by conjugation of *V. cholerae* co969 (recipient strain) with *E. coli* SM10 λ pir (pSC189) (*Chiang and Rubin, 2002*) (donor strain), as described by *Dörr et al., 2016*. Transposon mutants were selected on LB plates (10 g/l NaCl) containing both Sm(200 µg/ml) and Km (50 µg/ml). For the screen for biofilm repressors at 22°C, 459 out of 11,000 screened mutants were selected due to their rugose colony morphology at 22°C. To identify the genes interrupted by the transposon, the selected mutants were pooled together and transposon insertion sequencing (Tn-seq) was performed as described previously (*Chao et al., 2013*). In brief, the chromosomal DNA of the pooled mutants was purified and sheared by sonication (Covaris Sonicator E220). The overhanging ends were blunted using a blunting enzyme kit (NEB) and an A-tail was added with Taq DNA polymerase (NEB). Then, Illumina adaptors were ligated (with T4 DNA ligase, NEB) and the genomic DNA-transposon junctions were amplified (with Phusion High-Fidelity DNA polymerase, NEB). Sequencing was performed using an Illumina MiSeq benchtop sequencer (Illumina, San Diego, CA). Data analysis for determination of the TA dinucleotide insertion sites was conducted as described previously (*Chao et al., 2013*; *Pritchard et al., 2014*). For visualization of transposon insertion profiles, Sanger Artemis Genome Browser and Annotation tool were used (*Rutherford et al., 2000*).

To avoid selecting false-positive candidates with a temperature-independent effect on biofilm formation, another transposon mutagenesis was performed in parallel, by conjugation of the constitutively smooth *V. cholerae* C6706 with *E. coli* SM10 λ pir (pSC189). Out of 63,000 screened mutants, 618 formed rugose colonies at 37°C. Identification of the genes interrupted by the transposon was performed using the approach described above. Genes that were hit in both transposon screens were filtered out during the selection of candidates for further analysis. Higher number of reads per clone was used as a parameter for candidate selection.

A third transposon mutagenesis was performed by conjugation of *V. cholerae* co969 with *E. coli* SM10 λ pir (pSC189), for selection of mutants with smooth colony morphology at 37°C to screen for a potential BipA-specific protease. Out of 19,000 screened mutants, 313 were selected. Identification of the genes interrupted by the transposon was performed as described above.

The data for this study have been deposited in the ENA at EMBL-EBI under accession number PRJEB42487 (https://www.ebi.ac.uk/ena/browser/view/PRJEB42487).

## Western blotting of *V. cholerae* colonies

For Western blotting, two colonies were collected per sample, resuspended in buffer (20 mM Tris pH 7.5, 200 mM NaCl, 5 mM EDTA and protease inhibitor [cOmplete protease inhibitor cocktail tablets; Sigma Aldrich]), disrupted on a bead beater (Mini beadbeater, Biospec Products; 4°C, maximum speed for 1 min 15 s), and the total protein concentration of the samples was determined by Bradford assay (*Bradford, 1976*) with Protein Assay Dye Reagent Concentrate (Bio-Rad). Samples were normalized by total protein content and run in a 10% SDS-PAGE (sodium dodecyl sulfate polyacrylamide gel electrophoresis) gel for protein separation. Proteins were then transferred to an Immobilon-P transfer membrane (Millipore) using a Trans-Blot Turbo Transfer system (Bio-Rad). After blocking the membrane at room temperature for 2 hr with TBS-T buffer (24.2 g Trizma, 87.6 g NaCl in 1 l H$_2$O, pH 7.5, 0.1% (v/v) Tween-20) containing 5% (w/v) milk, the membrane was incubated at 4°C o/n with the monoclonal anti-flag M2 primary antibody (Sigma Aldrich) in TBS-T buffer containing 5% (w/v) milk, washed with TBS-T buffer, incubated at RT for 1 hr with α-mouse secondary antibody in TBS-T buffer and again washed with TBS-T buffer. BipA-flag was detected by addition of SuperSignal West Pico Chemiluminiscence substrate (Thermo Scientific) and detection of the signal using a LAS4000 (Fujifilm).

## Purification of BipA-His

*E. coli* BL21 cells carrying plasmid pET22b-*bipA*-His were grown at 37°C in LB media containing Cb (100 µg/ml) and 0.4% glucose (w/v) until $OD_{600}$ reached 0.4. Then, expression of the BipA-His fusion protein was induced by addition of 1 mM IPTG (isopropyl-β-d-thiogalactosidase) and further incubation for 2 hr at 37°C. Cells were then pelleted by centrifugation at 4°C 6000 rpm 30 min, resuspended in equilibration buffer (150 mM Tris-HCl pH 7.5, 200 mM NaCl) in the presence of protease inhibitors (cOmplete protease inhibitor cocktail tablets), and cell crude extracts were prepared by disruption in a pressure cell homogenizer FC600 (Julabo) followed by centrifugation at 4°C 15,000 rpm 30 min. BipA-His was purified from the soluble fraction by affinity chromatography using Ni-NTA agarose (Qiagen) previously equilibrated with equilibration buffer. After washing with washing buffer (150 mM Tris-HCl pH 7.5, 1 M NaCl), BipA-His was eluted with elution buffer (150 mM Tris-HCl pH 7.5, 200 mM NaCl, 500 mM imidazole) and then buffer exchanged to 100 mM Tris-HCl pH 7.5 + 100 mM NaCl by dialysis at 4°C o/n using a Spectra/Por molecular porous membrane tubing (6–8 kD) (Spectrumlabs.com). Purified proteins were stored either at 4°C, for immediate use, or at −80°C after addition of 10% glycerol. For CD experiments, an additional buffer exchange was performed to 20 mM sodium phosphate buffer pH 7.5, 50 mM NaCl by dialysis at 4°C o/n using a Spectra/Por molecular porous membrane tubing (6–8 kD) (Spectrumlabs.com).

## Circular dichroism

About 2.5 µM of BipA-His in 20 mM sodium phosphate buffer (pH 7.5, 50 mM NaCl) was pre-incubated for 30 min at the appropriate temperature (37, 22, or 15°C) or pre-incubated for 30 min at 37°C and then shifted to 22°C for further 30 min. Then, CD was measured in a Jasco J-720 CD spectrometer (Jasco, Japan; 190–260 nm; 1 mm quartz cuvette, 3 µm scan, five averaged scans). Background spectra were always subtracted prior to analysis.

## Sucrose gradient sedimentation

Cultures were grown in LB (10 g/l NaCl) media until mid-exponential phase. Cells were collected and lysed using a freeze–thaw method, and clarified lysates were analyzed by sucrose gradient sedimentation, as detailed previously (*Qin and Fredrick, 2013*), except that the ultracentrifugation run was extended by 1 hr. Absorbance at 254 nm was measured across the gradient, and data were quantified using Peak Chart (Brandel), software designed specifically for the ISCO/Brandel system employed. For each trace, areas under the 30S, 50S, 70S, and polysome peaks were integrated, and the corresponding values were normalized with respect to the 70S value.

## Quantitative label-free proteomics

Samples used for quantitative label-free proteomics were prepared as follows. Three biological replicates, each of them being a pool of four independent colonies of *V. cholerae* co969 or *V. cholerae* co969 Δ*bipA*, either grown at 37°C or 22°C (co969 37C, co969 22C, co969 Δ*bipA* 37C, co969 Δ*bipA* 22C) were analyzed. Total protein extracts were prepared by resuspending the collected pellets in 200 µl 20 mM Tris-HCl pH 7.5 + 200 mM NaCl, disrupting the cells in a bead beater (4°C, maximum speed for 1 min 15 s) and collecting the supernatants. Twenty micrograms of total protein extracts were run in a 12% pre-cast acrylamide gel at 100V for 20 min, and total protein bands were excised from the gel. Gel slices were digested as described previously (*Burian et al., 2015*). Peptide mixtures were then separated on an EasyLC nano-HPLC (Proxeon Biosystems) coupled to an LTQ Orbitrap Elite mass spectrometer (Thermo Fisher Scientific) as described elsewhere (*Carpy et al., 2014*) with the following modifications: peptides were eluted with an 130-min segmented gradient of 5–33–90% HPLC solvent B (80% acetonitrile in 0.5% acetic acid). The acquired MS spectra were processed with the MaxQuant software package, version 1.5.2.8 (*Cox and Mann, 2008*) with the integrated Andromeda search engine (*Cox et al., 2011*) as described previously (*Carpy et al., 2014*). Database searches were performed against a target-decoy *V. cholerae* complete database obtained from UniProt, containing 3783 protein entries and 248 commonly observed contaminants. The label-free algorithm was enabled, as was the 'match between runs' option (*Luber et al., 2010*). Label-free quantification protein intensities from the MaxQuant data output were used for relative protein quantification. Downstream bioinformatic analysis (Analysis of variance (ANOVA) and two-sample t-tests) was performed using the Perseus software package, version 1.5.0.15. $p < 0.05$ was considered

to be statistically significant. Determination of statistically significant differences was only possible for proteins that appeared in all four conditions.

### β-galactosidase assay

Samples for β-galactosidase assays were prepared as follows. Cultures of *V. cholerae* co969 WT or *V. cholerae* co969 Δ*bipA* carrying pCB192N derivatives with promoter-*lacZ* translational fusions were started from a 1:100 dilution of an o/n culture and grown to exponential phase. Two microliter drops were then plated on LB agar plates (10 g/l NaCl) + Cb (100 µg/ml) and incubated for an appropriate time at either 37°C or 22°C (under those conditions, the number of CFUs per colony was similar for both temperatures). For each sample, three to four colonies were collected and resuspended in buffer Z (60 mM $Na_2HPO_4.7H_2O$, 40 mM $NaH_2PO_4.H_2O$, 10 mM KCl, 1 mM $MgSO_4$, 50 mM β-mercaptoethanol, pH 7.0). β-galactosidase assays were performed as described previously (*Miller, 1972*). Miller units were calculated from two independent experiments containing three biological replicates each (each replicate containing four colonies).

### Statistical analyses

The program GraphPad PRISM Software (Inc, San Diego, CA, http://www.graphpad.com) has been used for all statistical analyses. To determine the significance of the data, the t-test (unpaired) has been performed. $p < 0.05$ has been considered significant.

## Acknowledgements

We thank Alonso R Serrano for assistance with RNA-seq data. We thank all the members of the Cava lab for helpful discussions. We thank Jörgen Ådén for assistance with CD experiments. Proteomics were performed by Ana Velic, Nicolas Nalpas, and Boris Mazek at the University of Tübingen (Germany). Research in the Cava lab is supported by The Swedish Research Council (VR), The Knut and Alice Wallenberg Foundation (KAW), The Laboratory of Molecular Infection Medicine Sweden (MIMS), and The Kempe Foundation. TdP was the recipient of an EMBO short-term fellowship (EMBO ASTF 1–2015). Research in the Waldor lab is supported by NIH grant RO1AI-042347 and HHMI. BS was supported by the Natural Sciences and Engineering Council of Canada (PGSD3-487259-2016). ARW was funded by grant T32 AI-132120. The work from the Fredrick lab was supported by NIH grant R01 GM072528.

## Additional information

### Funding

| Funder | Grant reference number | Author |
| --- | --- | --- |
| Swedish Research Council | | Felipe Cava |
| Knut och Alice Wallenbergs Stiftelse | | Felipe Cava |
| The Laboratory of Molecular Infection Medicine Sweden | | Felipe Cava |
| The Kempe Foundation | | Felipe Cava |
| EMBO | EMBO ASTF 1-2015 | Teresa del Peso Santos |
| National Institutes of Health | RO1AI-042347 | Matthew K Waldor |
| Natural Sciences and Engineering Research Council of Canada | PGSD3-487259-2016 | Brandon Sit |
| National Institutes of Health | T32 AI-132120 | Alyson R Warr |
| National Institutes of Health | R01 GM072528 | Kurt Fredrick |

The funders had no role in study design, data collection and interpretation, or the decision to submit the work for publication.

## Author contributions
Teresa del Peso Santos, Data curation, Formal analysis, Validation, Investigation, Visualization, Methodology, Writing - original draft, Writing - review and editing; Laura Alvarez, Oihane Irazoki, Benjamin R Warner, Kurt Fredrick, Formal analysis, Investigation, Visualization, Methodology, Writing - review and editing; Brandon Sit, Alyson R Warr, Anju Bala, Formal analysis, Investigation, Methodology, Writing - review and editing; Jonathon Blake, Vladimir Benes, Software, Formal analysis, Investigation, Methodology; Matthew K Waldor, Funding acquisition, Writing - original draft, Writing - review and editing; Felipe Cava, Conceptualization, Resources, Formal analysis, Supervision, Funding acquisition, Investigation, Visualization, Methodology, Writing - original draft, Project administration, Writing - review and editing

## Author ORCIDs
Laura Alvarez ![ORCID] https://orcid.org/0000-0003-2429-7542
Brandon Sit ![ORCID] http://orcid.org/0000-0003-2378-3039
Matthew K Waldor ![ORCID] http://orcid.org/0000-0003-1843-7000
Felipe Cava ![ORCID] https://orcid.org/0000-0001-5995-718X

## Decision letter and Author response
Decision letter https://doi.org/10.7554/eLife.60607.sa1
Author response https://doi.org/10.7554/eLife.60607.sa2

# Additional files

## Supplementary files
- Supplementary file 1. Supplementary figures 1–6.
- Supplementary file 2. Supplementary tables 1-3: strains, plasmids and primers used in this study.
- Supplementary file 3. RNA-seq.
- Supplementary file 4. RNA-seq – biofilm-related genes.
- Supplementary file 5. Transposon mutagenesis Vc co969 rugose colonies at 22℃.
- Supplementary file 6. Transposon mutagenesis Vc C6706 rugose colonies at 37℃.
- Supplementary file 7. Transposon mutagenesis Vc co969 smooth colonies at 37℃.
- Supplementary file 8. Proteomic raw data and analysis.
- Transparent reporting form

## Data availability
Sequencing data has been deposited in the European Nucleotide Archive (ENA) at EMBL-EBI under accession numbers PRJEB42487 and PRJEB42488. All data generated or analysed during this study are included in the manuscript and supplementary files.

The following datasets were generated:

| Author(s) | Year | Dataset title | Dataset URL | Database and Identifier |
| --- | --- | --- | --- | --- |
| del Peso T, Cava F | 2021 | Identification of determinants involved in temperature-dependent colony morphology in Vibrio cholerae co969 | https://www.ebi.ac.uk/ena/browser/view/PRJEB42487 | ENA, PRJEB42487 |
| del Peso T, Cava F | 2021 | RNA-Seq for identification of differentially expressed genes in Vibrio cholerae co969 colonies grown at different temperatures | https://www.ebi.ac.uk/ena/browser/view/PRJEB42488 | ENA, PRJEB42488 |

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
