## [Decision Letter]

**Acceptance summary:**

This work explores the molecular basis controlling the ability of *V. cholerae* strains to form smooth colonies at 22ºC and transition to rugose colonies at 37ºC. In this process, a ribosome-associated protein, named BipA, remodels the proteome at low temperature, modifying the levels of over 200 proteins. At 37ºC, in vitro experiments suggest that the levels of BipA decreases due to a conformational change that makes BipA more susceptible to protease degradation. Because BipA is a conserved GTPase, the findings could also be important to understand deep phenotypic transitions linked to temperature changes in other pathogens and bacteria in general.

**Decision letter after peer review:**

[Editors’ note: the authors submitted for reconsideration following the decision after peer review. What follows is the decision letter after the first round of review.]

Thank you for submitting your work entitled "A temperature-dependent translational switch controls biofilm development in *Vibrio cholerae*" for consideration by *eLife*. Your article has been reviewed by two peer reviewers, and the evaluation has been overseen by a Reviewing Editor and a Senior Editor. The following individual involved in review of your submission has agreed to reveal their identity: Jay Zhu (Reviewer #2).

Our decision has been reached after consultation among the reviewers. Based on these discussions and the individual reviews below, we regret to inform you that your work will not be considered further for publication in *eLife*.

As you will see from each individual review below, while the reviewers think that the work is sound and elegantly conducted, they both question whether BipA, being a translational factor, performs a more general task than controlling biofilm development in a temperature-dependent manner. After discussion, we decided that proving the specificity of BipA in temperature-dependent biofilm induction and thus, a bona fide BipA-dependent switch mechanism, would be required for publication in *eLife*. We hope that the reviewers comments will be helpful to prepare the manuscript for submission to another journal.

Reviewer #1:

The manuscript by del Peso et al., investigates the temperature-dependent regulation of smooth/rugose colony formation in *V. cholerae*. The results revealed that *V. cholerae* strains deficient in HapR, the master repressor of biofilm development, form smooth colonies at 22ºC and rugose colonies at 37ºC; because a protein named BipA represses the formation of rugose colonies at 22ºC. At 37ºC, in vitro experiments suggest that the levels of BipA decrease due to a conformational change that makes BipA more susceptible to protease degradation. BipA modifies the levels of over 200 proteins which are ultimately responsible for the rugose/smooth phenotype.

Overall, the manuscript has a solid logical flow. However, there are two main issues with the manuscript that require further clarification.

1) BipA is a ribosome associated GTPase. Thus, it is difficult to envision how this protein can be involved in a regulatory strategy for a specific group of proteins. BipA might be responsible for a basic function that affects the translation of proteins at low temperature, and the rugose phenotype of the colony might just be an indirect consequence of the absence of the protein. One simple experiment to investigate this possibility is to analyze the expression of a reporter protein (GFP, B-gal) in the WT and bipA mutant strains at 20ºC and 37ºC. If BipA is regulating the expression of some specific proteins, the levels of the reporter protein should not be affected by the absence of BipA.

2) In relation to the previous point, complementation of *V. cholerae* bipA mutant with bipA orthologues from *Escherichia coli* and *Pseudomonasaeruginosa* restore smooth colony morphology phenotype at 22ºC. I do not know how to interpret this finding. Do the authors think that these three bacteria share a regulatory pathway to control colony rugosity?

3) Another point that requires further clarification is the relationship between BipA and c-di-GMP. How does the absence of BipA affect c-di-GMP levels at 22ºC? Rugose colony morphology is usually related to high levels of c-di-GMP. Thus, the absence of BipA could cause an increase in the levels of c-di-GMP at 20ºC by favoring the accumulation of a diguanylate cyclase or reducing the levels of a phosphodiesterase.

4) Title. I think that the term "translational switch" is misleading. The levels of BipA seem to depend on a temperature-dependent structural conformation and susceptibility to protease activity. BipA modifies the levels of target proteins by a mechanism that remains unknown. Thus, it is not clear what the authors mean by translational switch.

5) Figure 1D (If I correctly understood the figure) suggests that a rugose colony grown at 37ºC contains smooth colonies, almost in similar numbers, to rugose colonies. Am I right? Does this mean that rugose colonies contain a mixture population of smooth and rugose bacteria? Is BipA mutated in these smooth colonies?

Reviewer #2:

The authors start with a nice phenotype (temperature-dependent rugose colony formation in certain *V. cholerae* strains); and through a series of painstaking experiments and screens, the authors discovered that a ribosome assembly factor BipA represses biofilm formation at low temperatures. They also provide data showing that BipA is less stable at 37C. The experimental designs are logical and data are nicely presented. The genetic screens are elegant. My major concern is that BipA has global effects on protein translation at low temperature and rugosity happens to be one of many. It is unclear what physiological role BipA plays and what mechanism of BipA regulation is (besides BipA stability). Major comments are listed below.

1) Rugose phenotype vs. biofilm formation. The authors are absolutely right that the rugose phenotype is linked to biofilm formation. However, since the temperature-dependent phenotype is new, the authors may want to perform "classical" biofilm assays in order to support their claim that temperature controls biofilm.

2) HapR issue. BipA effects are only observable in hapR- background (by the way, please offer an explanation why N16961 is an exception) but BipA stability is not affected by HapR. Thus the question is what the role of BipA is in hapR+ background? I wish the authors performed proteomic studies using a hapR+ background as well.

3) Growth of bipA mutants. The authors show that bipA does not affect growth at any temperature, but it seems that the growth condition tested (Supplementary figure 3) is different from growing on plates.

4) Figure 3C shows that at 37C, bipA- is no longer rugose. Isn't that BipA is a repressor and is not stable at 37C anyway? Similarly, in Figure 3D, no rugosity is seen at 37C.

5) BipA stability assays. The experiments are pretty crude for such an important aspect of the manuscript. It would be nice to see more quantitative measurements such as pulse-chase experiments. The authors may also want to show FLAG-tag itself does not affect protein stability and BipA-FLAG is functional. The trypsin partial digestion assays are poorly conducted. Even at 22C, with seconds, BipA-His has been degraded. We can argue that BipA is just very sensitive to trypsin digestion and the BSA control isn't.

---

## [Author Response]

[Editors’ note: The authors appealed the original decision. What follows is the authors’ response to the first round of review.]

Reviewer #1:The manuscript by del Peso et al., investigates the temperature dependent regulation of smooth/rugose colony formation in *V. cholerae*. The results revealed that *V. cholerae* strains deficient in HapR, the master repressor of biofilm development, form smooth colonies at 22ºC and rugose colonies at 37ºC; because a protein named BipA represses the formation of rugose colonies at 22ºC. At 37ºC, in vitro experiments suggest that the levels of BipA decrease due to a conformational change that makes BipA more susceptible to protease degradation. BipA modifies the levels of over 200 proteins which are ultimately responsible for the rugose/smooth phenotype.Overall, the manuscript has a solid logical flow. However, there are two main issues with the manuscript that require further clarification.1) BipA is a ribosome associated GTPase. Thus, it is difficult to envision how this protein can be involved in a regulatory strategy for a specific group of proteins. BipA might be responsible for a basic function that affects the translation of proteins at low temperature, and the rugose phenotype of the colony might just be an indirect consequence of the absence of the protein. One simple experiment to investigate this possibility is to analyze the expression of a reporter protein (GFP, B-gal) in the WT and bipA mutant strains at 20ºC and 37ºC. If BipA is regulating the expression of some specific proteins, the levels of the reporter protein should not be affected by the absence of BipA.

We agree with this comment insofar as BipA regulates the levels of proteins beyond those involved in biofilm/rugose colony formation. However, we believe our proteomics data suggest that BipA is not a global translational regulator. To further investigate this possibility, we performed two additional experiments. First, we constructed additional translational reporters in housekeeping proteins predicted to be unaffected by BipA from the proteomics, and indeed found that the levels of these proteins do not change between low or high temperatures (Figure 6D in the revised manuscript), whereas there are substantial shifts in biofilm-associated putative BipA-impacted proteins like VpsR and VpsL.

We also performed additional polysome analysis on lysates from WT and *bipA*-mutant co969 *V. cholerae* and found that there is a modest defect in 50S subunit assembly in the mutant strain (Figure 5 in the revised manuscript). A similar defect has been reported in *E. coli* to also preferentially alter the levels of some, but not all proteins (Gibbs et al., 2020), supporting the idea that BipAdependent control of ribosome assembly has targeted effects on the proteome.

We recognize that the previous version of the manuscript portrayed BipA as a biofilmspecific regulator, which was not our intention. We have significantly re-written parts of the manuscript to more accurately convey our conclusion, which is that BipA does indeed alter the levels of proteins responsible for the phenotype of interest (colony rugosity/biofilm formation), but also influences proteins thought to participate in a wide variety of cellular functions.

2) In relation to the previous point, complementation of *V. cholerae* bipA mutant with bipA orthologues from *Escherichia coli* and *Pseudomonas aeruginosa* restore smooth colony morphology phenotype at 22ºC. I do not know how to interpret this finding. Do the authors think that these three bacteria share a regulatory pathway to control colony rugosity?

Since the BipA orthologues we tested in this study share high sequence identity, it is not surprising that complementation of the *V. cholerae bipA* mutant with these sequences rescued the inappropriate 22C rugosity phenotype, as it suggests a functional BipA in the context of the *V. cholerae* biofilm regulatory network exerts temperature-dependent control of colony rugosity. However, since we did not characterize *E. coli* or *P. putida* BipA mutants, we cannot conclude that this specific function of BipA in those species, which have biofilm regulatory schemes that differ significantly from *V. cholerae*, is the same. We think it is likely BipA controls the levels of proteins in those species’ proteomes, but without extensive further work whether its specific impact on biofilm-associated proteins and downstream colony rugosity is conserved remains an open question.

3) Another point that requires further clarification is the relationship between BipA and c-di-GMP. How does the absence of BipA affect c-di-GMP levels at 22ºC? Rugose colony morphology is usually related to high levels of c-di-GMP. Thus, the absence of BipA could cause an increase in the levels of c-di-GMP at 20ºC by favoring the accumulation of a diguanylate cyclase or reducing the levels of a phosphodiesterase.

To address this point, we compared the levels of c-di-GMP for the WT and *bipA* mutant strains at both 37ºC and 22ºC (Author response image 1). We did not observe significant differences in the levels of this second messenger between the two strains at either temperature, indicating that c-di-GMP is not involved in the BipA-mediated control of colony rugosity. We note that known diguanylate cyclases in the *V. cholerae* genome were unaltered in the comparative RNAseq analysis in Figure 2, further suggesting that this phenotype is c-di-GMP-independent.

**Author response image 1. sa2fig1:** c-di-GMP quantification in *V. cholerae* WT and Δ*bipA* mutant at different temperatures. A) Representative HPLC chromatograms of *V. cholerae* soluble nucleotide profiles. The peak corresponding to c-di-GMP peak is marked with an arrow and elutes with similar retention time as the commercial c-di-GMP standard used as control. B) Relative amount of c-di-GMP normalized to total protein content in *V. cholerae* co969 and Δ*bipA* mutant colonies incubated at 37°C and 22°C at different timepoints.

4) Title. I think that the term "translational switch" is misleading. The levels of BipA seem to depend on a temperature-dependent structural conformation and susceptibility to protease activity. BipA modifies the levels of target proteins by a mechanism that remains unknown. Thus, it is not clear what the authors mean by translational switch.

This is a fair point, and we agree that our data do not conclusively identify the mechanism by which BipA modifies the levels of certain proteins in the proteome, only that this process is temperature dependent and possibly a result of a defect in ribosome subunit assembly in the mutant strains. We have revised our title to “BipA exerts temperature-dependent control of biofilm-associated colony morphology in *Vibrio cholerae*” and appropriately scaled back claims regarding BipA’s effect on translation throughout the manuscript.

5) Figure 1D (If I correctly understood the figure) suggests that a rugose colony grown at 37ºC contains smooth colonies, almost in similar numbers, to rugose colonies. Am I right? Does this mean that rugose colonies contain a mixture population of smooth and rugose bacteria? Is BipA mutated in these smooth colonies?

Figure 1D does not show mixtures of smooth and rugose colonies – rather, it shows the colony-forming units (i.e. amount of viable bacteria) in colonies collected from conditions where smooth or rugose colonies are present. Since CFU densities did not vary greatly between smooth and rugose colonies, we concluded that this process was occurring independently of cell growth (i.e. the increased contrast of the colony is not due to simple increases in cell number).

Reviewer #2:The authors start with a nice phenotype (temperature-dependent rugose colony formation in certain *V. cholerae* strains); and through a series of painstaking experiments and screens, the authors discovered that a ribosome assembly factor BipA represses biofilm formation at low temperatures. They also provide data showing that BipA is less stable at 37C. The experimental designs are logical and data are nicely presented. The genetic screens are elegant. My major concern is that BipA has global effects on protein translation at low temperature and rugosity happens to be one of many. It is unclear what physiological role BipA plays and what mechanism of BipA regulation is (besides BipA stability). Major comments are listed below.1) Rugose phenotype vs. biofilm formation. The authors are absolutely right that the rugose phenotype is linked to biofilm formation. However, since the temperature dependent phenotype is new, the authors may want to perform "classical" biofilm assays in order to support their claim that temperature controls biofilm.

The reviewer makes a fair point that rugose colony morphology is not a direct readout of biofilm formation. However, we would like to emphasize that colony morphology was not the only biofilm “readout” we used in this study. The RNAseq, qPCR and proteomic analyses all invariably identified strong upregulation of biofilm-associated structural and regulatory genes in rugose colonies compared to smooth colonies.

To address this point, we revised the manuscript to refrain from making direct statements about biofilm formation except for in the Discussion. In the revised manuscript, we now refer to colony morphology more specifically, as well as to “biofilm-associated” phenotypes to more accurately reflect our data. We also changed the title of the manuscript to “BipA exerts temperature-dependent control of biofilmassociated colony morphology in *Vibrio cholerae*”.

2) HapR issue. BipA effects are only observable in hapR- background (by the way, please offer an explanation why N16961 is an exception) but BipA stability is not affected by HapR. Thus the question is what the role of BipA is in hapR+ background? I wish the authors performed proteomic studies using a hapR+ background as well.

To address the reviewer’s first point, N16961 may be an exception to our identification of BipA effects in HapR- strains due to additional genetic background changes that prevent this strain from forming rugose colonies at high temperatures. N16961 has over 200 SNPs compared to other commonly used strains such as C6706 (and likely co969), and dissecting this observation, while biologically interesting, is beyond the scope of this paper.

The role of BipA in HapR+ background is an interesting point for discussion. In the revised manuscript, we are careful to emphasize that BipA does not solely impact biofilm gene regulation. As shown by our proteomics studies, BipA influences the levels of many other proteins with roles in diverse cellular processes such as motility, cell envelope maintenance and metabolism. We verified that at least one of these processes (motility) is indeed altered in *bipA* mutant strains, and that this phenotype is in fact independent of HapR (Supplementary figure 7). Thus, the HapR dependence of BipA’s effect on colony rugosity may be a special circumstance. Since HapR levels are regulated by external inputs, specifically quorum sensing, BipA could plausibly still impact biofilm formation and/or colony morphology in HapR+ strains in environments where HapR is intact, but not produced. We suspect that performing the proteomics in a HapR+ background would be interesting, but ultimately yield many of the same hits that the HapR- analyses revealed. In other words, BipA would impact the abundance of the same proteins, but in the case of HapR+ strains, the ultimate downstream influence on biofilm formation would be masked by the dominant effects of HapR (leaving BipA’s effect on other cell processes intact).

3) Growth of bipA mutants. The authors show that bipA does not affect growth at any temperature, but it seems that the growth condition tested (Supplementary figure 3) is different from growing on plates.

We performed additional imaging of plate-grown *V. cholerae* to address this point. As shown in Supplementary figure 3, there is no visible difference between WT and BipAdeficient strains in colony growth at different times or different temperatures.

4) Figure 3C shows that at 37C, bipA- is no longer rugose. Isn't that BipA is a repressor and is not stable at 37C anyway? Similarly, in Figure 3D, no rugosity is seen at 37C.

BipA- colonies are more rugose than WT colonies and have higher contrast values, and hence seem darker in the representative images. We suspect this may be due to minimal or barely detectable levels of BipA activity at 37C, so in the complete absence of this protein there is indeed a subtle increase in rugosity at this temperature.

5) BipA stability assays. The experiments are pretty crude for such an important aspect of the manuscript. It would be nice to see more quantitative measurements such as pulse-chase experiments. The authors may also want to show FLAG-tag itself does not affect protein stability and BipA-FLAG is functional.

The reviewer makes a valid point that the stability assays are not high-resolution snapshots of BipA stability. However, we note that the primary observation in the manuscript in this section is that BipA protein levels are drastically changed at different temperatures, while *bipA* transcript levels remain the same. We have re-written this section to reflect that temperature-mediated sensitivity to proteolysis is just one hypothesis for a model of control of BipA’s abundance, and that our current data do not allow us to conclusively determine this mechanism.

As the reviewer requested, we generated a stable co969 strain expressing only the BipA-FLAG fusion protein. This strain exhibits identical colony morphology to the WT strain (Author response image 2), strongly suggesting that the fusion protein is functional and not drastically altered in stability or abundance.

**Author response image 2. sa2fig2:** Colony morphology of BipA-FLAG derivative.

The trypsin partial digestion assays are poorly conducted. Even at 22C, with seconds, BipA-His has been degraded. We can argue that BipA is just very sensitive to trypsin digestion and the BSA control isn't.

We agree with the reviewer that the trypsin proteolysis data indicates BipA is exquisitely sensitive to trypsin digestion compared to a control protein such as BSA and this assay is not suitable to address this question. Hence, we have decided to remove these results from the revised manuscript.